dilated cardiomyopathy; heart failure; cardiovascular genetics; precision medicine; gene therapy

**Corresponding author:**
Brian P. Halliday;
Email: b.halliday@imperial.ac.uk

# Precision therapy in dilated cardiomyopathy: Pipedream or paradigm shift?

Saad Javed[1,2] and Brian P. Halliday[1,2] (iD)

[1]National Heart and Lung Institute, Imperial College London, UK and [2]Cardiovascular Research Centre, Cardiovascular Magnetic Resonance Unit & Inherited Cardiac Conditions Care Group, Royal Brompton and Harefield Hospitals, Part of Guy's and St Thomas' NHS Foundation Trust, London, UK

## Abstract

Precision medicine for cardiomyopathies holds great promise to improve patient outcomes costs by shifting the focus to patient-specific treatment decisions, maximising the use of therapies most likely to lead to benefit and minimising unnecessary intervention. Dilated cardiomyopathy (DCM), characterised by left ventricular dilatation and impairment, is a major cause of heart failure globally. Advances in genomic medicine have increased our understanding of the genetic architecture of DCM. Understanding the functional implications of genetic variation to reveal genotype-specific disease mechanisms is the subject of intense investigation, with advanced cardiac imaging and mutliomics approaches playing important roles. This may lead to increasing use of novel, targeted therapy. Individualised treatment and risk stratification is however made more complex by the modifying effects of common genetic variation and acquired environmental factors that help explain the variable expressivity of rare genetic variants and gene elusive disease. The next frontier must be expanding work into early disease to understand the mechanisms that drive disease expression, so that the focus can be placed on disease prevention rather than management of later symptomatic disease. Overcoming these challenges holds the key to enabling a paradigm shift in care from the management of symptomatic heart failure to prevention of disease.

## Impact statement

Advances in the understanding of the molecular mechanisms that cause dilated cardiomyopathy offer the opportunity to personalise care and improve the outcomes of patients with this heterogeneous family of disease. Comprehensive characterisation of the disease with genetic testing and advanced imaging will play a key role. Precision therapies that target the primary disease mechanism will offer new hope for disease prevention in genetically susceptible individuals at risk of developing highly penetrant, malignant forms of the condition as well as effective treatments of early asymptomatic disease.

## Introduction

Heart failure is a looming global health crisis with a predicted lifetime risk of 25 to 45% that is rapidly reaching epidemic proportions (Huffman et al., 2013; Benjamin et al., 2019). Despite the already high risk of developing heart failure, current projections indicate that the prevalence of the condition will surge by 46%, and treatment expenditure will increase by a staggering 127% by 2030 (Heidenreich et al., 2013; Huffman et al., 2013). These sobering statistics call for a radical shift in our current approach to managing the disease.

Dilated cardiomyopathy (DCM) is a myocardial disorder characterised by left ventricular (LV) dilatation accompanied by systolic dysfunction, in the absence of abnormal loading conditions or coronary artery disease (Yancy et al., 2013; Pinto et al., 2016; Heidenreich et al., 2022). Its prevalence is around 1 in 220 people and it represents the leading indication for heart transplantation (Japp et al., 2016; Chambers et al., 2018). DCM arises from a range of genetic and acquired factors, often occurring simultaneously. There is a significant overlap between intrinsic and extrinsic causes (Figure 1). Conditions that were previously considered as separate aetiologies, such as peripartum cardiomyopathy, cardiomyopathy following anthracycline chemotherapy and alcohol-related cardiomyopathy have been shown to have similar genetic backgrounds (Ware et al., 2016; Ware et al., 2018; Garcia-Pavia et al., 2019). It is therefore perhaps best to consider DCM as a family of related disease that require comprehensive geno- and phenotyping to fully understand (Yancy et al., 2013; Japp et al., 2016; Pinto et al., 2016).

Present treatment strategies centre on the management of symptomatic heart failure using guideline-directed heart failure management (GDMT) – a combination of beta-blockers,



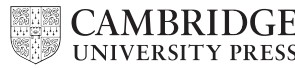

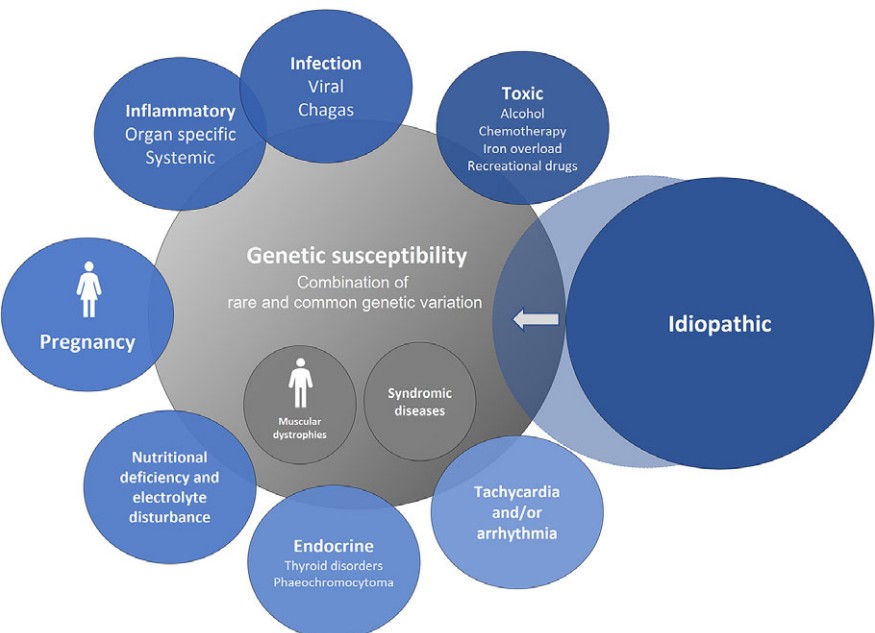

**Figure 1.** Dilated cardiomyopathy (DCM) – a family of diseases. Selected syndromic causes of dilated cardiomyopathy include Barth syndrome, haemochromatosis, Kearns–Sayre syndrome, and Carvajal syndrome (adapted with permission from Halliday, 2022).

angiotensin-converting enzyme inhibitors, mineralocorticoid antagonists and SGLT-2 inhibitors (Japp et al., 2016; McDonagh et al., 2021; Heidenreich et al., 2022). This easily generalisable approach has dramatically improved the outcome of patients with heart failure reduced ejection fraction over the last 40 years (Vaduganathan et al., 2020). However, it places little focus on early treatment of asymptomatic myocardial dysfunction before the onset of heart failure and instead predominantly targets the neurohormonal consequences of the heart failure syndrome. The question of the "right time" to start these treatments is unclear and typically these agents are commenced when patients develop symptoms, late in the disease pathway. An increasing number of asymptomatic individuals with mild disease or genetically susceptible individuals are being identified through screening strategies. The incorporation of genetic information into DCM care offers opportunities for early precision intervention in individuals at risk before they develop symptoms (Figure 2). Whilst current GDMT will undoubtedly continue to form a mainstay of symptomatic heart failure, precision medicine offers a revolutionary solution that could not only offer additional targeted therapies for those with more advanced disease but perhaps, more importantly, offer targeted therapies to prevent and slow disease expression much earlier in the disease course. Currently, there is limited evidence focusing on the treatment of early DCM. This is likely to be related to low event rates, a long latent period in the development of overt disease and variability in the natural history of different gene mutations. However, with increasing numbers of individuals being identified at risk, it is important that streamlined clinical trials across large populations, using pragmatic end-points set out to address this important issue.

The fundamental premise is that a nuanced understanding of an individual's disease through advanced cardiac imaging, genetics, and biomarkers enables a more personalised understanding of the mechanism and guides a more refined and targeted therapeutic approach. But is this all just a pipedream, or are we really on the brink of a paradigm shift in DCM care? In this article, we examine the present state and future promise of precision medicine in DCM

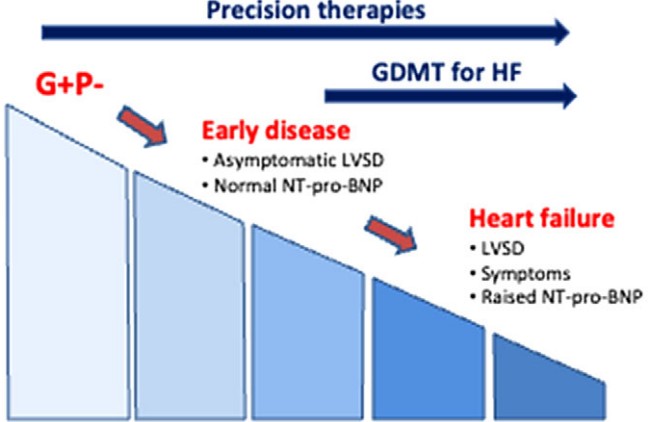

**Figure 2.** Precision therapies for genotype-positive, phenotype-negative (G+ P−) individuals would likely involve genotype-specific therapies, and lifestyle interventions. Treatments that could be introduced at an early disease stage include anti-fibrotic agents, and therapies to target cardiac metabolism (such as SGLT2 inhibitors) whereas advanced disease therapies include antiarrhythmics for those at the highest risk, ICD therapy and guideline-directed heart failure therapy (GDMT) (HF, heart failure; LVSD, left ventricular systolic dysfunction).

to guide novel treatments using precision phenotyping of disease. We provide an overview of existing knowledge regarding the genetic origins of DCM and contemporary approaches to individualised management.

## The genetic architecture of DCM

Using currently available next-generation sequencing panels, a causative rare genetic variant in about 20–30% of cases of DCM (Tayal et al., 2021). The yield may be higher in select populations, such as those referred for advanced heart failure therapies, younger patients or those with a high burden of ventricular arrhythmia or conduction disease (Herman et al., 2012; Lukas Laws et al., 2022).

Several different genes encoding a range of proteins with diverse functions have been implicated in DCM (Jordan et al., 2021). These are most commonly autosomal with X-linked and mitochondrial variants also uncommonly identified. Autosomal dominant inheritance is the norm and the most common genes implicated are those coding for sarcomeric proteins, including titin (*TTN*) and beta-myosin (*MYH7*). Other notable genes include those coding for cytoskeletal (*FLNC*), nuclear envelope (*LMNA*) and desmosomal (*DSP*) proteins. In some cases, the identification of a causative variant may influence treatment decisions, particularly those associated with a more malignant prognosis. However, the greatest value of genetic testing comes in predicting the risk of asymptomatic relatives by enabling the identification of those at high risk of developing the disease in the future. In gene elusive disease, all first-degree family relatives typically remain under clinical surveillance until the age of 60 years. If a causative genetic variant is identified, cascade screening will be able to identify the 50% of relatives who are carriers and who have an elevated risk of developing disease. Those who do not carry the variant can be discharged from follow-up.

A landmark study found rare truncating variants in the titin (*TTNtv*) gene in 25% of patients with advanced or familial DCM (Herman et al., 2012). More recent studies found *TTNtv* in 13% of nonfamilial cases of DCM and ~ 0.5% of the general population (McNally and Mestroni, 2017; Tayal and Prasad, 2018; Schultheiss et al., 2019; Verdonschot et al., 2019). These variants are associated with incomplete penetrance and variable expression (Japp et al., 2016) that may be attributed to additional gene-modifying factors, environmental factors, or the penetrance of disease later in life (Japp et al., 2016). Penetrance is likely to vary significantly between asymptomatic carriers with a family history of DCM and those found to carry a *TTNtv* as a secondary finding on testing performed for another reason. Nevertheless, subtle markers of reduced cardiac function have been found in carriers in the general population (Schafer et al., 2016), suggesting they may be more susceptible to extrinsic insults, such as alcohol and cardiotoxic chemotherapy.

Recent research has emphasised the importance of common genetic variation or polygenic risk in determining the risk of developing DCM (Pirruccello et al., 2020; Tadros et al., 2021). Many patients without a rare variant cause are likely to have high polygenic risk contributing to the development of contractile dysfunction along with extrinsic factors. Polygenic risk is also likely to influence the penetrance of rare genetic variants, helping to explain the variable expressivity and incomplete penetrance commonly seen across families with a pathogenic variant. A future precision approach to determining disease risk in families with DCM is likely to integrate data on phenotype, rare and common genetic variation and the interaction with extrinsic insults.

Autosomal recessive transmission has also been described. This is of particular relevance in younger individuals and childhood cardiomyopathies. For example, biallelic loss of function mutations in the nebulin-related anchoring protein gene (*NRAP*) have been identified in some individuals with severe sporadic DCM and have been proposed to cause low-penetrant recessive disease (Iuso et al., 2018; Koskenvuo et al., 2021). Several syndromic causes of DCM have been identified. These include dystrophinopathies such as Duchenne and Becker's muscular dystrophy, and other eponymous syndromes including Barth syndrome (Hershberger et al., 2009; Hershberger et al., 2013). A raised creatinine kinase level and characteristic sub-epicardial fibrosis in the lateral wall are typical in patients with dystrophinopathies and cardiac manifestations may predate neuromuscular symptoms in Becker's muscular dystrophy (Del Rio-Pertuz et al., 2022). Rare metabolic disorders, particularly inborn errors of metabolism, have also been associated with DCM (Guertl et al., 2000; Cox, 2007). Broadly, these can be grouped into disorders of amino acid/organic acid metabolism, disorders of fatty acid metabolism, glycogen and lysosomal storage disorder and mitochondrial disorders (Guertl et al., 2000).

## Stratifying arrhythmic risk in genetic DCM

The traditional approach to stratifying the risk of major ventricular arrhythmia in DCM relies on a combination of symptoms and left ventricular ejection fraction (LVEF). However, this "cause-agnostic" approach does not fully encapsulate the heterogeneity of DCM and a growing body of data support an increased risk of SCD with specific genotypes. This has begun to influence international guidelines on the selection of patients for implantable cardioverter defibrillators (ICDs) that recommend lower thresholds for such devices in patients with *LMNA, FLNC, PLN* or *RBM20* variants and other high-risk features beyond LVEF (Zeppenfeld et al., 2022). This represents a wider trend in recent guidelines that attempt to risk stratify patients according to genotype and phenotype to make personalised decisions about their care, attempting to break down the traditional grouping of "non-ischemic cardiomyopathy" (Table 1).

## Is genotype-specific therapy the answer?

Discovering a monogenic cause for DCM provides direct insight into the molecular mechanisms that drive disease. This creates the possibility of using precision therapies directed at the primary molecular basis of disease. Such approaches are not only relevant to those with symptomatic heart failure where they may be used alongside GDMT, but perhaps more importantly for asymptomatic individuals with early markers of disease or those with genetic susceptibility to developing disease later in life. Evidence-based treatments for the latter groups are currently lacking. Targeted therapy for disease prevention must be a priority. Strategies targeting the primary disease mechanisms may take different main approaches.

The immediately downstream molecular consequences of the variant represent attractive targets. Most genes associated with cardiomyopathy serve important functions within the cardiomyocyte, with their respective proteins carrying out specific functions. Disruptions in the function of these proteins, either through loss or gain of function, result in intracellular changes in signal transduction, prompting the cardiomyocyte to undergo adaptive changes (Reichart et al., 2019). Given the heterogeneity of DCM, downstream targeting of these processes would require a wide variety of agents to target the products of different genes implicated in the pathogenesis. One such target that was recently investigated in phase II and III trials was the heightened cardiac activity of ERK1/2, JNK, and p38 MAP kinases downstream from variants in *LMNA* associated with DCM (Muchir et al., 2012). Much hope arose from animal studies that demonstrated a reduction in adverse remodelling following the administration of a p38 inhibitor (Wu et al., 2011; Laurini et al., 2018). Unfortunately, these results were not translated into the phase III trial that was recently stopped due to futility.

Another example comes from the use of myosin modulators in sarcomeric DCM. Sarcomeric dysfunction is the primary mechanism of DCM in patients with *TTNtv* or relevant variants in *MYH7*. This is the opposite functional consequence of sarcomeric variants

**Table 1.** Genes with definite/strong association with DCM and their functional and phenotypic implications (14, 17)

| Gene | Protein | Function | Phenotype/risk |
|---|---|---|---|
| TTN | Titin | Extensible scaffold/molecular spring | Low prevalence of LBBB, atrial fibrillation |
| LMNA | Prelamin-A/C | Nuclear membrane structure | Accelerated disease Ventricular arrhythmia may precede overt DCM |
| FLNC | Filamin-C | Structural integrity of cardiac myocyte; actin crosslinking protein | Ventricular arrhythmia may precede overt DCM, Overlapping phenotype of dilated and left-dominant arrhythmogenic cardiomyopathies complicated by frequent premature SCD |
| RBM20 | RNA-binding motif protein 20 | Regulates cardiac gene splicing | High risk of sudden cardiac death, Malignant VAs |
| PLN | Cardiac phospholambin | Sarcoplasmic reticulum calcium regulator, inhibits SERCA2a pump | Founder mutation in Netherlands, high risk of SCD, Significant posterolateral and free wall fibrosis in PLN R14del |
| SCN5A | Sodium channel protein type-5 subunit alpha | Sodium channel | Ventricular arrhythmia may proceed overt LV dysfunction |
| DSP | Desmoplakin | Desmosomal junction protein | Ventricular arrhythmia risk, Extensive fibrosis may precede LV systolic dysfunction and LV dilatation |
| BAG3 | BAG family molecular chaperone regulator 3 | Inhibits apoptosis | High penetrance > 40 years, worse prognosis in nonsense variants |
| TNNTC1 | Troponin C, slow skeletal and cardiac muscles | Myocardial contraction | |
| TNNT2 | Troponin T, cardiac muscle (troponin T2) | Myocardial contraction | |

causing hypertrophic cardiomyopathy (HCM) that are associated with sarcomeric over-action. In the same way that promise has arisen from the use of myosin inhibitors, there is excitement about the potential use of agents such as danicamtiv and omecamtiv mecarbil that increase actin-myosin cross-bridging in sarcomeric DCM (Voors et al., 2020; Teerlink et al., 2021). Similarly, emerging data suggest that *TTNtv* are associated with modifications in cardiac metabolism and energy utilisation (Verdonschot et al., 2018; Ware and Cook, 2018; Zhou et al., 2019). In particular, an upregulation in the transcription of important mitochondrial machinery may represent a compensatory response to sarcomeric dysfunction (Ware and Cook, 2018). Targeting early mitochondrial dysfunction may therefore be a promising target for future investigation.

An example of precision therapy from current clinical practice is the use of sodium channel blockers such as flecainide or quinidine for DCM associated with *SCN5A* variants that result in an increased sodium current (Peters et al., 2022). A recent systematic review has shown such cardiomyopathies, typically associated with a high burden of ventricular arrhythmias, to be responsive to sodium channel blockers (Peters et al., 2022). It may be argued that such phenotypes are a primary electrical disease rather than a true cardiomyopathy. Nevertheless, the reversibility with widely available therapies emphasises the importance of achieving a genetic diagnosis, avoiding other unnecessary invasive procedures (Figure 3).

Arguably, the most definitive treatment approaches are those that directly target the genetic variant (Verdonschot et al., 2019).

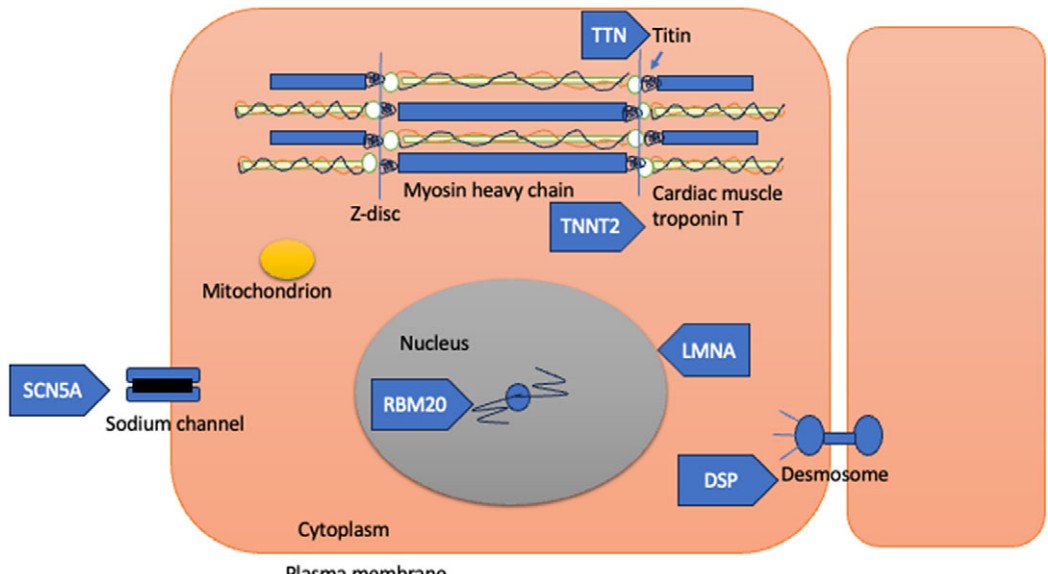

**Figure 3.** Cellular locations of some of the proteins with their respective genes associated with dilated cardiomyopathy.

Various methods are currently being investigated to accomplish this objective, including: (1) gene editing – the use of CRISPR/Cas9 to directly edit the genetic sequence and restore normal protein function, (2) gene replacement therapy for cardiomyopathies associated with loss of function variants where the wild type gene is expressed, primarily through gene transfer techniques with viral vectors, (3) gene silencing therapy, primarily using small interfering RNA molecules to reduce the expression of abnormal functioning protein as a result of missense variants, and (4) exon skipping, involving the use of anti-sense oligonucleotides to mask exons during transcription and restoring the reading frame (Carrier et al., 2015; Gramlich et al., 2015; Prondzynski et al., 2017; Ma et al., 2018).

Much of the early progress in this area has been in Duchenne muscular dystrophy (DMD) where both exon skipping and gene editing have been used to restore dystrophin production in experimental models (Amoasii et al., 2018). Early work has also demonstrated the potential of similar approaches in DCM associated with *TTN,* another similarly large gene with areas of redundant sequence (Gramlich et al., 2015; Romano et al., 2022). Pre-clinical studies in mice with the well-described *PLN* R14 gene deletion, have also used anti-sense oligonucleotides to decrease phospholamban activity, prevent cardiac dysfunction, and improve survival (Grote Beverborg et al., 2021; Deiman et al., 2022). This particular founder variant is associated with a malignant form of DCM, commonly encountered in the Netherlands.

Although substantial progress has been made in demonstrating the feasibility and potential of genome editing in cellular and murine models, numerous unanswered questions remain prior to advancing to human trials involving currently available techniques. Key considerations include ensuring the safety of viral delivery and accurately targeting the vector to the intended site with appropriate dosage (Colella et al., 2018).

### Precision therapy in gene elusive disease

Whilst genetic therapies hold great promise for those with monogenic disease, the majority will have little relevance for the majority of patients with DCM without a rare variant genetic cause. This group of patients are likely to have a diverse range of disease mechanisms including activation of fibroinflammatory pathways and metabolic dysfunction, driven by extrinsic causes including toxic insults, inflammatory or metabolic disease as well as genetic susceptibility related to common genetic variation (Reichart et al., 2022). Characterising these mechanisms in individual patients using precision phenotyping may help guide targeted therapy. The integration of advanced cardiac imaging and biomarkers offer huge potential to individualise management.

### *Myocardial fibrosis*

In DCM the balance between extracellular matrix (ECM) synthesis and degradation is disrupted (Piek et al., 2016). This leads to the formation myocardial fibrosis. Fibrosis is initiated by the activation and differentiation of fibroblasts into myofibroblasts, triggered by transforming growth factor (TGF-β) (Khalil et al., 2017). Myofibroblasts produce higher levels of ECM proteins, contributing to the development of fibrosis (Nagaraju et al., 2019). Fibrosis leads to reduction in compliance of the diseased myocardium and acts as a substrate for arrhythmias (de Jong S et al., 2011; Ellims et al., 2014). It is recognised as a key disease mechanism across a spectrum of

DCM and is thought to represent a modifiable target for treatment, particularly in early disease before replacement fibrosis or scar has developed (Halliday and Prasad, 2019).

Fibrosis is likely to be driven via multiple different pathways. Neurohormonal activation as part of the heart failure syndrome with upregulation of angiotensin II and aldosterone is likely to play an important role (Halliday and Prasad, 2019). Myocardial inflammation and immune activation are also tightly linked to fibrotic pathways and are likely to play an important role in a subset of patients (Halliday and Prasad, 2019). Upregulation of fibrosis also appears to be an early feature of specific genotypes including *FLNC, DSP* and *LMNA* (Augusto et al., 2020). Targeting patients in these groups with anti-fibrotic agents may therefore be fruitful.

Mineralocorticoid receptor antagonists, which are an important part of GDMT show promise as potential antifibrotic drugs for patients with DCM (Izawa et al., 2005; Al-Khatib et al., 2018; McDonagh et al., 2021). These medications can influence remodelling, reduce biomarkers associated with collagen biosynthesis, and improve patient outcomes (Sharma et al., 2004; Besler et al., 2017). Evidence also suggests that antifibrotic agents used in other diseases, such as pirfenidone, may hold some promise in the treatment of heart failure (Lewis et al., 2021).

### *Cardiac metabolism*

A key characteristic of DCM and heart failure is reduced oxidative metabolism and a shift from fatty acid oxidation to increased glucose utilisation (Heggermont et al., 2016). Whether this is adaptive or maladaptive remains a topic of debate. Other important metabolic changes include increased ketone metabolism that is thought to represent a therapeutic target. Regardless of the cause, a myocardial energy deficit appears to be an important pathway in perpetuating the progression of the disease (Heggermont et al., 2016; Sacchetto et al., 2019).

It appears likely that the myocardial energetic phenotype and impact of impaired myocardial energetics will differ across the spectrum of DCM. This may be influenced by co-morbidities such as diabetes mellitus as well as age that are associated with impairment of energetics (Chowdhary et al., 2022). Genotype-specific differences are also likely to exist. In recent studies, the impact of DCM-causing *TTNtv* was explored in rats, revealing a correlation with impaired autophagy, reduced oxygen consumption rate, increased production of reactive oxygen species (ROS), and elevated ubiquitination of mitochondrial proteins in cardiomyocytes (Sacchetto et al., 2019; Zhou et al., 2019). This is supported by data from human myocardial tissue demonstrating important changes in the transcription of proteins relevant to mitochondrial function in carriers of *TTNtv* (Verdonschot et al., 2018; Reichart et al., 2022). Additionally, an aberrant signalling pathway involving ERK1/2 was associated with altered mitochondrial shape, distribution, fragmentation, and degeneration in a mouse model of *LMNA* DCM (Galata et al., 2018).

There are many possible metabolic modulators that could be studied in a targeted fashion. There is interest in the use of the antioxidant and cofactor for mitochondrial electron transport, coenzyme Q10. Phase III trial data in heart failure with reduced ejection fraction was promising, however larger, more robust trials are required before routine clinical use (Mortensen et al., 2014). A mitochondrial-targeted form of coenzyme Q10, MitoQ, has also gained interest following convincing experimental data (Goh et al., 2019). Whether some forms of DCM, such as those related to TTNtv, may gain more benefit from such therapies is unclear. It

is also possible that such therapies will improve cardiac function through other pathways, such as by reducing endothelial dysfunction and reducing afterload (Roura and Bayes-Genis, 2009; Giannitsi et al., 2019). Trimetazidine inhibits the protein thiolase I, responsible for the final step of beta-oxidation in the mitochondria. This results in a shift in substrate utilisation towards glucose metabolism (Tuunanen et al., 2008). Perhexilene reduces fatty acid oxidation by inhibiting carnitine palmitoyltransferase-1 and similarly promotes a switch to glucose utilisation (Beadle et al., 2015). Early phase data have suggested that such agents may improve myocardial energetics and LV systolic function, however, later phase data are still lacking and concerns regarding the long-term safety of perhexiline remain (Tuunanen et al., 2008; Zhang et al., 2012; Beadle et al., 2015; Fan et al., 2018). Debate continues whether downregulating fatty oxidation is truly beneficial (Watson et al., 2023). Much therefore remains to be understood about the role of personalised metabolic therapy.

Given the likely variable impact of fibrosis, immune activation and metabolic dysfunction across the spectrum of DCM, it is essential that we have accessible non-invasive methods to assess the role of these mechanisms in individual cases to guide precision and targeted therapies. Cardiac imaging as well as circulating biomarkers have the potential to play an important role.

## Cardiac imaging

Whilst echocardiography (TTE) serves as the initial modality for diagnosing patients with heart failure with reduced ejection fraction, it is unable to reliably discriminate the cause of left ventricular dysfunction. Much data supports the use of cardiac magnetic resonance (CMR) imaging as a valuable tool for discriminating between ischaemic and non-ischaemic aetiologies and refining the cause and mechanism of non-ischaemic LV dysfunction (Japp et al., 2016; Halliday, 2022). It does so through detailed tissue characterisation using late gadolinium enhancement (LGE) imaging and parametric mapping (Japp et al., 2016; Halliday, 2022; Merlo et al., 2023). This insight currently provides important information that guides selection of patients for ICDs and may also help individualise other treatment decisions in the future.

LGE represents replacement myocardial fibrosis and is present in around one-third to a half of cases (Kuruvilla et al., 2014; Di Marco et al., 2017). LGE presence has been found to be a predictor of mortality, hospitalisation, and sudden cardiac death (SCD). Furthermore, the presence, extent, and patterns of LGE may provide additional valuable predictive information regarding malignant ventricular arrhythmias (VAs) or left ventricular (LV) reverse remodelling (Kuruvilla et al., 2014). The presence of LGE is now included in guidelines for primary prevention ICD implantation (McDonagh et al., 2021; Zeppenfeld et al., 2022). The pattern of myocardial fibrosis on CMR may also point towards particularly genetic aetiologies. Variants in desmoplakin (*DSP*) and filamin C (*FLNC*) have been shown to be associated with ring-like patterns of myocardial fibrosis which has been associated with worse outcomes (Augusto et al., 2020). Parametric mapping with CMR also offers the ability to quantify interstitial changes, including fibrosis and oedema. Another exciting emerging fibrosis imaging technique is 68-gallium-labelled fibroblast activation protein inhibitor (FAPI) positron emission tomography (PET). This nuclear technique offers the potential to image fibrosis activity, anticipate fibrotic remodelling and prevent clinical disease before it occurs using targeted anti-fibrotic therapies.

[31]Phosphorus magnetic resonance offers the unique ability to study myocardial energetics in vivo. Studies have confirmed that DCM is characterised by a decrease in the ratio of phosphocreatine to adenosine triphosphate, a marker of impaired energetics (Stoll et al., 2016). This has been shown to improve with reverse remodelling and predict outcome (Neubauer et al., 1997). This technique offers the ability to characterise the metabolic phenotype of individual patients and perhaps identify those who may gain most benefit from targeted metabolic therapies.

Diffusion tensor CMR enables comprehensive evaluation of cardiac microstructure revealing intricate details of myocardial wall mechanics, including the rotational torsion of myocardial sheetlets. This emerging technique may offer unique insight into the response to therapies targeting the sarcomere (Nielles-Vallespin et al., 2017).

## Blood biomarkers

Circulating biomarkers provide the opportunity to characterise metabolic derangement, collagen turnover as well inflammatory and immune activation (Rubis et al., 2022). This has the potential to guide therapy decisions. One potential disadvantage is that many are not cardiac-specific. For example, circulating serum biomarkers of fibrosis reflect collagen turnover not only in the heart but also in various organs such as vessels, liver, and bone. Nevertheless, the carboxy-terminal propeptide of procollagen type I (PICP) and the amino-terminal propeptide of procollagen type III (PIIINP) have been correlated cardiac fibrosis observed on histology (Izawa et al., 2005; Lopez et al., 2010; Rubis et al., 2022) and elevated levels of these peptides predict an unfavourable outcome in patients with HF (Martos et al., 2009; Sweeney et al., 2020; Cleland et al., 2021). There has been interest in using markers to select patients who may benefit the most from anti-fibrotic therapy (Cleland et al., 2021; Raafs et al., 2021). Galectin-3 is another marker of fibro-inflammatory activity and has been identified as a prognostic marker due to its association with worse outcomes in DCM (Sharma et al., 2004; Besler et al., 2017). It appears likely that fibrosis plays an important role in driving early disease in particular phenotypes. The extent to which biomarkers will be able to guide therapy prior to the emergence of symptomatic DCM is unknown. One advantage of using them in susceptible individuals or those with early disease is that extra-cardiac causes of fibrosis are less likely to be relevant in this younger, less co-morbid group. Additionally, other markers such has high-sensitivity troponin T (hsTnT) and N-terminal prohormone brain natriuretic peptide (nt-proBNP) may have an important role in predicting disease progression (Chmielewski et al., 2020; Suresh et al., 2022). Both, for example, have been associated with the risk of malignant ventricular arrythmias in LMNA mutation carriers (Figure 4).

## Precision phenotyping in DCM

Another key challenge is integrating these multidimensional data in a simple, accessible way to create a ground truth for the patient we see in clinic. Several studies have used unbiased clustering analysis known as phenomapping, in patients with various forms of heart failure including DCM, to help define subgroups of patients (Shah et al., 2015; Verdonschot et al., 2020; Tayal et al., 2022). The heterogenous aetiology of DCM makes it imminently suitable for this form of classification. Tayal et colleagues used a machine-learning based approach to cluster patients based on

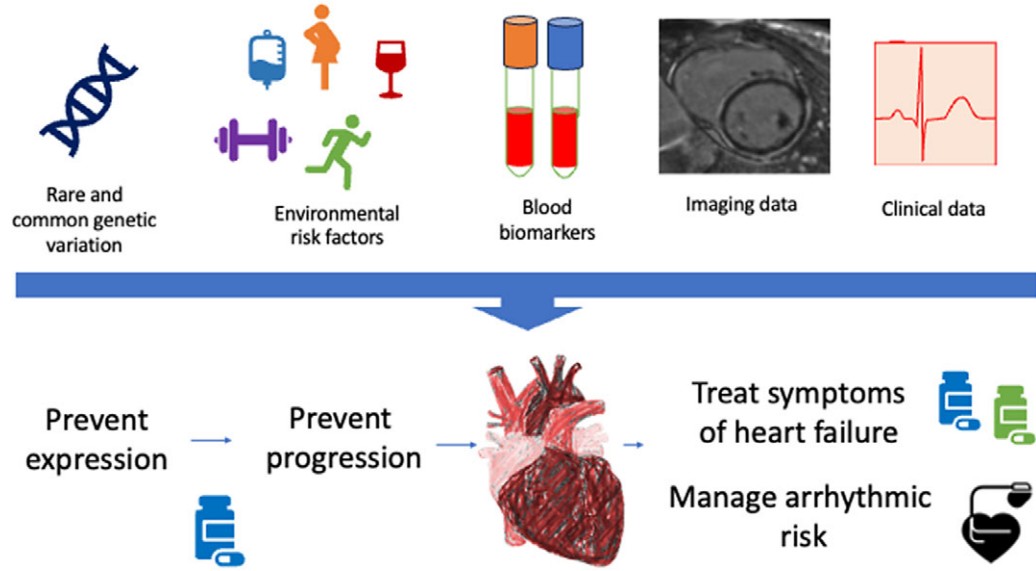

**Figure 4.** Precision medicine for dilated cardiomyopathy.

clinical, imaging, genetic and circulating characteristics and identified distinct subclasses of DCM with shared and distinct disease mechanisms (Tayal et al., 2022). Verdonschot and colleagues used a similar approach incorporating transcriptomics to identify distinct transcriptomic profiles, including, pro-fibrotic, pro-inflammatory and metabolic subtypes (Verdonschot et al., 2020). Both groups then used common clinical variables to discriminate between the groups so that this approach could be translated more easily into clinical practice.

By untangling the upstream causes and downstream active processes unique to each patient, such approaches may illuminate the targets for therapeutic intervention. The heterogeneity of DCM necessitates a personalised approach, with treatment strategies designed to benefit the individual patient subgroups that emerge from thorough phenotypic characterisation.

## Co-morbidities and lifestyle

In the individualised treatment of individuals with DCM, it is important to also manage comorbidities such as coronary artery disease, hypertension, diabetes, thyroid disease, anaemia, and obesity (Reichart et al., 2019; Verdonschot et al., 2019; Zeppenfeld et al., 2022). It is likely that such co-morbidities interact with intrinsic susceptibility to develop contractile impairment. Whether more intensive treatment and stricter control of these issues improves outcomes remains unclear. Special attention should also be paid to the impact of alcohol and cardiotoxic chemotherapy, such as anthracyclines (Ware et al., 2018; Andersson et al., 2022; Tayal et al., 2022). Whilst it is clear that excessive amounts of alcohol may be harmful, it is debatable whether low or moderate levels of consumption lead to adverse remodelling and unclear whether abstinence should be recommended (Andersson et al., 2022). It is possible that specific genotypes may lead to increased susceptibility to cardiotoxins (Ware et al. 2018). Individualised exercise prescription is another important factor to consider. Patients with symptomatic DCM or features of increased risk should avoid engaging in high-intensity or competitive sports (Pelliccia et al., 2019). There is particular concern for those with high-risk genotypes.

## Conclusion

A precision medicine approach holds great promise for revolutionising our approach to patients with the heterogeneous family of diseases that make up DCM. By integrating findings from clinical data, genetic testing, advanced imaging and circulating biomarkers, clinicians can gain a detailed understanding of each patient's disease that can help individualise treatment via a shared decision-making approach.

However, significant challenges remain. Integrating the breadth of available genomic and phenotypic data to predict individual risk remains a challenge. Whilst many disease-specific treatments are under investigation, some remain years away from clinical routine. Whilst disease mechanisms have been well characterised in advanced disease, at what stage these occur in the natural history of DCM and whether early targeted intervention will delay the onset of overt disease remains to be determined. Despite these hurdles, the incorporation of genomic and phenotypic data hold the potential to establish a novel clinical framework for evidence-based and personalised care in DCM.

**Open peer review.** To view the open peer review materials for this article, please visit http://doi.org/10.1017/pcm.2023.24.

**Data availability statement.** Data sharing not applicable – no new data generated.

**Author contribution.** Both authors contributed to the literature search, data analysis and manuscript preparation.

**Financial support.** B.P.H. is supported by a BHF Intermediate Clinical Research Fellowship awarded to B.P.H. (FS/ICRF/21/26019) and the Rosetrees Trust.

**Competing interest.** B.P.H. has served on an advisory board for Astra Zeneca. S.J. declares no competing interest.

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
