## [Reviewer Report]

Javed and Halliday presented a very insightful, informative and well balanced review on the current knowledge on background and potential methods of precision therapy in dilated cardiomyopathy (DCM). Indeed, a very interesting, well composed approach.

Whether and what more to take into consideration?

In the paragraph 2, Authors omitted autosomal recessive transmission that may occur, especially in younger more severe affected subjects, both with homozygotes (like NRAP) and compound heterozygotes. Rare metabolic causes, e.g. type IV glycogenosis, especially in pediatric/adolescent population could also be taken into consideration. Furthermore, syndromic forms, with other striated muscle disease, related mainly to dystrophinopathies could be mentioned, and the role of CK assessment, helpful in the diagnostic proces, could be stressed.

Apart from additional role of common genetic variation in determining the risk of developing DCM, it seems relevant to unveil the role of susceptibility variants, often marked as VUS-es in known genes associated with DCM, e.g. RBM20. As it is known in TAAD patients ( PMID: 29961567), similarly VUSes in the genes associated with DCM can be low penetrant „risk variants”, in particular in the setting of acute onset heart failure.

In the paragraph 3 I wonder whether endothelial dysfunction should not be addressed.

In the paragraph 5 refering to biomarkers, the role of hs troponins in the early stage of cardiomyopathies could be shown (PMID: 32408651).

On page 14 maybe it is not so clear, with regard to Verdonschot publication (2020). Authors identified 4 phenogroups (mild systolic dysfunction, auto immune, cardiac arrhythmia, severe systolic dysfunction) and 3 transcriptomic profiles (pro-inflammatory, pro-fibrotic, metabolic).

---

## [Editor Report]

Congratulations to the authors. This paper has approached the multi paradigms that are present in the current context of dilated cardiomyopathy, and for the next few years. This theme is wide, but the authors have carried out a dense revision on this limited space to the manuscript. Please consider the following corrections:

In figure 1. There is a blue circle under the “idiopathic circle” which has no title. What does this blue circle represent, in fact? You should describe it in the figure. Further, this figure can be better with an explanation on what “syndromic causes” (cited in the picture) means. You can write about it at the figure footer.

Figure 2: What means G+P- should be described in the figure footer, as well as LVSD, GDMT and HF, although these terms could be identified in the text. You should also cite examples of what precision therapies can be applied in each phase of the disease progression. 

Page 7, line 21: If a causative genetic variant is identified, cascade screening will be able to identify the 50% of relatives who are carriers and who have an elevated risk of developing disease.

I think that you are considering 50% the risk of heritable transmission associated with autosomal dominant disease; however, in some pedigrees, all the offspring (100%) or nobody (0%) have inherited the causal variant. This comment seems to be also wrong in X-linked, recessive or mitochondrial etiologies where 50% is an error. Please, remove the 50% in the sentence.

Page 7, line 45: Nevertheless, subtle markers of reduced cardiac function have been found in carriers in the general population(Schafer et al. 2016), suggesting they may be more susceptible to extrinsic insults. 

Please, cite what are the extrinsic insults.

Page 8, line 15. You should explain LVEF, because it was the first time that you cite this abbreviation in the text.

---

## [Editor Report]

Dear, 

We are very grateful for the modifications made by you authors. However, there is one review that is not correct:

>>> Page 7, line 21: If a causative genetic variant is identified, cascade screening will be able to identify the 50% of relatives who are carriers and who have an elevated risk of developing disease.

>>> I think that you are considering 50% the risk of heritable transmission associated with autosomal dominant disease; however, in some pedigrees, all the offspring (100%) or nobody (0%) have inherited the causal variant. This comment seems to be also wrong in X-linked, recessive or mitochondrial etiologies where 50% is an error. Please remove the 50% in the sentence.

>>> Page 6, line 10. The text has been modified to read: “If a causative rare variant in an autosomal gene is identified, cascade testing will be able to identify 50% of the relatives who are carriers and who have an elevated risk of developing diseases.

As we had mentioned previously. Saying that it is able to identify 50% of relatives who are carriers is a misinterpretation of a dominant autosomal risk. Please include the sentence as follows:

“If a causative genetic variant is identified, cascade screening will be able to identify the 50% of relatives who are carriers and who have an elevated risk of developing disease.”

Best regards

---

## [Editor Report]

Congratulations on your review manuscript. We hope that it can help physicians and other precision medicine professionals to do the best care to DCM patients.

Best regards